# Comparing Supervised Machine Learning Strategies and Linguistic Features to Search for Very Negative Opinions

**Sattam Almatarneh *** and **Pablo Gamallo**

Centro Singular de Investigación en Tecnoloxías da Información (CITIUS),
Universidad de Santiago de Compostela, Rua de Jenaro de la Fuente Domínguez,
15782 Santiago de Compostela, Spain; pablo.gamallo@usc.es
**\*** Correspondence: sattam.almatarneh@usc.es; Tel.: +34-631648949

**Abstract:** In this paper, we examine the performance of several classifiers in the process of searching for very negative opinions. More precisely, we do an empirical study that analyzes the influence of three types of linguistic features (n-grams, word embeddings, and polarity lexicons) and their combinations when they are used to feed different supervised machine learning classifiers: Naive Bayes (NB), Decision Tree (DT), and Support Vector Machine (SVM). The experiments we have carried out show that SVM clearly outperforms NB and DT in all datasets by taking into account all features individually as well as their combinations.

**Keywords:** sentiment analysis; opinion mining; linguistic features; classification; very negative opinions

## 1. Introduction

The era of mass information is the most prominent feature of this century. The world has been transformed into a small village interconnected with the increase of social networking sites where it is possible that anyone anywhere on the planet can sell, buy or express their opinions. The enormous amount of information on the Internet has become an important source of data for different pieces of work and studies, as it offers the opportunity to extract information and organize it according to particular needs.

From the massive use of the Internet and social media in different aspects of life, social media has begun to play a crucial role in guiding people's trends in a multitude of fields: religious, political, social and economic—all this through the opinions expressed by individuals.

Sentiment Analysis also called Opinion Mining is defined as the field of study that analyzes people's opinions, sentiments, evaluations, attitudes, and emotions from written language. It is one of the most active research areas in Natural Language Processing (NLP) and is also widely studied in data mining, Web mining, and text mining [1].

Sentiment analysis typically works at four different levels of granularity, namely document level, sentence level, aspect level, and concept level. Most early studies in Sentiment Analysis [2,3] put their focus at a document level and relied on datasets such as movie and product reviews. After Internet use becoming widespread and the e-commerce boom, different types of datasets have been collected from websites about customer opinions. The review document often expresses opinions on a single product or service and was written by a single reviewer.

According to Pang et al. [4], between 73% and 87% readers of online reviews such as hotels, restaurants, or travel agencies state that reviews have a significant influence on their purchase.

The fundamental task in Opinion Mining is the classification of polarity [4–6], which consists of classifying a text by means of a predefined set of polarity categories: for example, positive, neutral,

negative. Reviews such as "thumbs up" versus "thumbs down", or "like" versus "dislike" are examples of bipolar polarity classification between two equally probable classes. However, it is possible to make other types of classifications. For example, a more unusual way of conducting sentimental analysis is to detect and classify opinions that represent the most negative opinions on a subject, object, or individual. We call them extreme opinions.

The most negative opinion is the worst view, judgment, or appraisal formed in one's mind about a particular matter. People always want to know the worst aspects of goods, services, places, etc. so that they can avoid them or fix them. Very negative opinions have a strong impact on product sales, as they have a great influence on customers' decisions before buying. Previous studies analyzed this correlation, such as the experiments described in [7], which show that as the proportion of negative online consumer feedback increases, so do the negative attitudes of consumers in general. Similar effects have also been seen in consumer reviews: single-star reviews hurt book sales on Amazon.com [8] enormously. It should also be borne in mind that the impact of one-star reviews, which represent the most negative opinions, is greater than the impact of five-star reviews (the most positive ones) on this trade sector.

The main objective of this article is to examine the effectiveness and limitations of different linguistic features and supervised sentiment classifiers to identify the most negative opinions in four domains' reviews. It is an expanded version of a conference paper presented at KESW 2017 [9]. Our main contribution is to report on a broad set of experiments aimed at evaluating the effectiveness of different linguistic features in a supervised classification task. In addition, we also compare some supervised classifiers, namely Support Vector Machine (SVM), Naive Bayes (NB), and Decision Tree (DT), for a binary classification task consisting of searching for very negative vs. not very negative opinions.

The rest of the paper is organized as follows. In the following Section 2, we discuss the related work. Then, Section 3 describes the method. Experiments are introduced in Section 4, where we also describe the evaluation and discuss the results. We draw the conclusions and future work in Section 5.

## 2. Related Work

In related work, we find two main approaches to search for the polarity of sentiments at the document level. First, machine learning techniques based on training corpora annotated with polarity information and, second, strategies based on polarity lexicons.

In machine learning, there are two main methods, unsupervised and supervised learning, even though only the later strategy is used by most existing techniques for document-level sentiment classification. Supervised learning approaches use labeled training documents based on automatic text classification. A labeled training set with a pre-defined categories is required. A classification model is built to predict the document class on the basis of pre-defined categories. The success of supervised learning mainly depends on the choice and extraction of the proper set of features used to identify sentiments. There are many types of classifiers for sentiment classification using supervised learning algorithms:

- Probabilistic classifiers such as Naive Bayes (NB), Bayesian networks and Maximum Entropy (ME).
- Decision tree classifiers, which build a hierarchical tree-like structure with true/false queries based on categorization of training documents.
- Linear classifiers, which separate input vectors into classes using linear (hyperplane) decision boundaries. The most popular linear classifiers are SVM and neural networks (NN).

One of the pioneer pieces of research on document-level sentiment analysis was conducted by Pang et al. [3] using probabilistic methods (NB and ME) as well as linear classifiers (SVM) for binary sentiment classification of movie reviews. They also tested different features, to find out that SVM with unigrams yielded the highest accuracy.

SVM is one of the most popular supervised classification methods. It has a solid theoretical basis, is likely the most precise method in text classification [10] and is also successful in sentiment classification [11–13]. It generally outperforms NB and finds the optimal hyperplane to divide classes [14]. Moraes et al. [15] compared NB and SVM with NN for sentiment classification. Experiments were performed on both balanced and unbalanced dataset. For this purpose, four datasets were chosen, namely movies review dataset [16] and three different product reviews (Cameras, Books and GPS). For unbalanced datasets, the performance of the two classifiers, SVM and NN, were affected in a negative way. Bilal et al. [17] compared the efficiency of three techniques, namely NB, Nearest Neighbour and Decision Tree, in order to classify both English and Urdu opinions of blogs. Their results showed that NB performs better than the other two classifiers. Table 1 summarizes the main components of some published studies: techniques utilized, the granularity of the analysis (sentence-level or document-level, etc.), type of data, source of data, and language.

**Table 1.** Main components of some supervised learning sentiment classification published studies.

| Ref. | Techniques Utilized | Granularity | Type of Data | Language |
|---|---|---|---|---|
| [3] | ME, NB, SVM | Document level | Movie reviews | English |
| [18] | MNB [1], ME, SVM | Sentence level | Blog, Review and News forum | English, Dutch, French |
| [12] | SVM | Document level | Movie, Hotel, Products | English |
| [19] | NB, ME, SVM | Document level | Movie, Products | English |
| [20] | Rule-based, SVM | Sentence level | Movie, Products | English |
| [15] | SVM, NN | Document level | Movie, GPS, products | English |
| [21] | NB, SVM, KNN [2] | Document level | Education, sports, political news | Arabic |
| [22] | ME, SVM | Document level | Movie, Products | Czech |
| [23] | NB | Sentence level | Products | English |
| [24] | SVM | Document level | Products | English, Italian |
| [25] | NN | Aspect level | Hotel | English |

[1] Multinomial Naive Bayes: is a specialized version of Naive Bayes that is designed more for text documents.
[2] k-nearest neighbors.

The quality of the selected features is a key factor in increasing the efficiency of the classifier for determining the target. Some of the most typical features are n-grams, word embedding, and sentiment words. All of them have been employed in different research on sentiment analysis. The influence in the classification task of these content features has been evaluated and analyzed by some opinion mining studies [3,26,27].

Tripathy et al. [28] proposed an approach to find the polarity of reviews by converting text into numeric matrices using countvectorizer and, Term Frequency-Inverse Document Frequency (TF-IDF), and then using it as input in machine learning algorithms for classification. Martín-Valdivia et al. [29] combined supervised and unsupervised approaches to obtain a meta-classifier. Term Frequency (TF), Term Frequency-Inverse Document Frequency (TF-IDF), Binary Occurrence (BO), and Term Occurrence (TO) were considered as feature representation schemes. TF-IDF was reported as the better representation scheme. SVM using TF-IDF weight scheme for unigrams without stopwords but with a stemmer yielded the best precision outperforming NB classifier. Paltoglou and Thelwall [30] examined several unigram weighting schemes. They found that some variants of TF-IDF are the best schemes for Sentiment Analysis.

Sentiment words, also called opinion or polarity words, are considered the primary building block in sentiment analysis as they represent an essential resource for most sentiment analysis algorithms. They are seen as the first indicator to express positive or negative opinions. There are, at least, two ways of creating sentiment lexicons: hand-craft construction based on lexicographic guidelines [31–34],

and automatic strategy on the basis of external resources. Two different automatic methods can be identified according to the nature of these resources: structured thesaurus and unstructured text corpora.

Goeuriot et al. [35] described the creation of two corpus-based lexicons. First, a general lexicon generated using SentiwordNet and the Subjectivity Lexicon. Second, a domain-specific lexicon created using a corpus of drug reviews depending on statistical information. Mohammad and Turney [36] elaborated a lexicon consisting of a combination of polarity (positive, negative) along with one of eight possible emotion classes for each word. The emotion classes involved in this work are: anger, anticipation, disgust, fear, joy, sadness, surprise, and trust.

Unlike our previous work [9,37], to our knowledge, no other work has focused on the automatic detection of very negative opinions. Therefore, our proposal might be considered as the first attempt in that direction.

## 3. Method

In this section, we will describe the most important linguistic features and supervised sentiment classifiers that we will use in our experiments.

We have focused on the selection of influential linguistic features taking into account the importance of the quality of the selection of features as a key factor in increasing the efficiency of the classifier in determining the target. The main linguistic features we will use and analyze are the following: N-grams, word embeddings, and sentiment lexicons.

### 3.1. N-Gram Features

First, we model texts by n-grams based on the occurrence of both unigrams and migraphs of words that occur in documents. The unigrams (1g) and bigrams (2g) are very valuable elements to find very relevant expressions in the domain of interest, in this case in the domain of polarity or opinion.

All terms are assigned a weight by means of two different quantitative representations: TF-IDF and CountVectorizer.

TF-IDF is computed in Equation (1):

$$tf/idf_{t,d} = (1 + log(tf_{t,d})) \times log(\frac{N}{df_t}),\tag{1}$$

where $tf_{t,d}$ is the term frequency of term $t$ in document $d$. $N$ stands for the the number of documents in the collection and, $df_t$ represents the number of documents in the collection containing $t$.

CountVectorizer converts the documents into a matrix with token counts. The process is the following. First, the documents are tokenized and a sparse matrix is created according to the number of occurrences of each token. To build the matrix, we remove all stopwords from the document collection. Then, we clean up the vocabulary by filtering out those terms appearing in less than four documents. This way, all terms that are too infrequent are removed.

In order to transform the reviews into the two matrices: a matrix of TF-IDF features and another one of token occurrences, we used *sklearn* feature extraction Python library. (http://scikit-learn.org/stable/modules/generated/sklearn.feature_extraction.text.CountVectorizer.html) (http://scikit-learn.org/stable/modules/generated/sklearn.feature_extraction.text.TfidfVectorizer.html#sklearn.feature_extraction.text.TfidfVectorizer)

### 3.2. Word Embeddings

Many deep learning strategies in NLP require word embeddings, i.e., vector spaces based on distributional semantics, as input features. Word embeddings can be considered as a technique for language modeling and feature learning that converts the words of a vocabulary into vectors of continuous real numbers representing their contextual distribution. This technique commonly

involves embeddings derived from a high-dimensional sparse vector space which is transformed into a lower-dimensional and dense vector space. Each dimension of the dense vector stands for a latent feature of a word. Dense or embedding vectors try to encode linguistic regularities that represent linguistic patterns of the word contexts. The process of learning word embeddings may be done by means of neural networks.

To represent the reviews, we make use of the *doc2vec* algorithm described in Le and Mikolov [38]. This neural-based model has been shown to be efficient when you have to account for high-dimensional and sparse data [38,39]. Doc2vec learns corpus features using an unsupervised strategy and provides a fixed-length feature vector as output. The output is then fed into a machine learning classifier. We used a freely available implementation of the doc2vec algorithm included in gensim, (https://radimrehurek.com/gensim/) which is a free Python library. The implementation of the doc2vec algorithm requires the number of features to be returned (length of the vector). Thus, we performed a grid search over the fixed vector length 100 [40–42].

## 3.3. Sentiment Lexicons

Sentiment words, also called opinion or polarity words, are considered the main component of sentiment analysis, as they are an essential resource for most sentiment analysis algorithms, as well as the most typical element to provide positive or negative opinions. In addition, many textual features may be used to detect very negative views. In this study, we have extracted some of them to examine to what extent they influence the identification of extreme views (very negative ones). Uppercase characters may indicate that the writer is very upset, so we counted the number of words written in uppercase letters. In addition, intensifier words could be a reliable indicator of the existence of very negative views. Thus, we considered words such as mostly, hardly, almost, fairly, really, completely, definitely, absolutely, highly, awfully, extremely, amazingly, fully, and so on.

Furthermore, we took into account negation words such as *no, not, none, nobody, nothing, neither, nowhere, never, etc. In addition, we also considered elongated words and repeated punctuation such as sooooo, baaaaad, woooow, gooood, ???, !!!!*, etc. These textual features have been shown to be effective in many studies related to polarity classification such as Taboada et al. [31], Kennedy and Inkpen [43]. In our previous studies [44,45], we described a strategy to build sentiment lexicons from corpora. In the current study, we will use our lexicon, called VERY-NEG (https://github.com/almatarneh/LEXICONS) which contains a list of very negative words (VN) and a list of words that are not considered to be very negative (NVN). VERY-NEG lexicon was built from the text corpora described in Potts [46]. The corpora (http://www.stanford.edu/~cgpotts/data/wordnetscales/) consist of online reviews collected from IMDB, Goodreads, OpenTable and Amazon/Tripadvisor. Each of the reviews in this collection has an associated star rating: one star (very negative) to ten stars (very positive) in IMDB, and one star (very negative) to five stars (very positive) in the other online reviews.

Reviews were tagged using the Stanford Log-Linear Part-Of-Speech Tagger. Then, tags were broken down into WordNet PoS Tags: *a* (adjective), *n* (noun), *v* (verb), *r* (adverb). Words whose tags were not part of those categories were filtered out. The list of selected words was then stemmed.

Table 2 summarizes all the features introduced above with a brief description for each one.

**Table 2.** Description of all linguistic features.

| Features | Descriptions |
|---|---|
| N-grams | Unigram TF-IDF (1g) <br> Unigram CountVectorizer (1g) <br> Unigram and Bigram TF-IDF (1g 2g) <br> Unigram and Bigram CountVectorizer (1g 2g) |
| Doc2Vec (100 Feat.) | Generate vectors for the document |
| Lexicons (12 feat.) | Number and proportion of VN terms in the documents <br> Number and proportion of NVN terms in the documents <br> Number and proportion of negation words in the document <br> Number and proportion of uppercase words in the document <br> Number and proportion of elongated words and punctuation in the document <br> Number and proportion of intensifiers words in the document |

## 4. Experiments

### 4.1. Multi-Domain Sentiment Dataset

This dataset (https://www.cs.jhu.edu/~mdredze/datasets/sentiment/index2.html) was used in Blitzer et al. [47]. It contains product reviews taken from Amazon.com for four types of products (domains): Kitchen, Books, DVDs, and Electronics. The star ratings of the reviews are from 1 to 5 stars. In our experiments, we adopted the scale with five categories. In this case, the borderline separating the VN values from the rest was set to 1, which stands for the very negative reviews. The documents in the other four categories were put in the not very negative (NVN) class.

### 4.2. Hotel Dataset

In order to extract extreme opinions, we require to analyze document collections with scaled opinion levels (e.g., rating) and extract those documents associated with the lowest and highest scale. We obtained our dataset from Expedia crowdsourced data. The HotelExpedia dataset (http://ave. dee.isep.ipp.pt/~1080560/ExpediaDataSet.7z) originally contains 6030 hotels and 381,941 reviews from 11 different hotel locations. The datasets are cleaned and prepared for analysis by applying the following three preprocessing steps: (1) data deduplication operation is performed in order to remove such duplicate reviews; (2) 3-star reviews were deleted since they tend to contain neutral views; and (3) all reviews containing less than three words and blank reviews were also removed. After the three data cleansing operations above, the final datasets consists of 20,000 reviews, being 5000 for each category: 1, 2, 4 and 5 stars. Table 3 shows the number of reviews in each class for each task.

**Table 3.** Size of the four test datasets and the total number of reviews in each class negative vs. positive and (VN vs. NVN).

| Datasets | # of Reviews | Negative | Positive | VN | NVN |
|---|---|---|---|---|---|
| **Books** | 2000 | 1000 | 1000 | 532 | 1462 |
| **DVDs** | 2000 | 1000 | 1000 | 530 | 1470 |
| **Electronics** | 2000 | 1000 | 1000 | 666 | 1334 |
| **Kitchens** | 2000 | 1000 | 1000 | 687 | 1313 |
| **Hotel** | 20,000 | 10,000 | 10,000 | 5000 | 15,000 |

### 4.3. Training and Test

Since we are facing a text classification problem, any existing supervised learning methods can be applied. Support Vector Machines (SVM), Naive Bayes (NB), and Decision Tree (DT) have been shown to be highly effective at traditional text categorization [3]. We decided to utilize *scikit* (http: //scikit-learn.org/stable/), which is an open source machine learning library for Python programming

language [48]. We chose SVM, NB and DT as our classifiers for all experiments; hence, in this study we will compare, summarize and discuss the behaviour of these learning models with the linguistic features introduced above. Supervised classification requires two samples of documents: training and testing. The training sample will be used to learn various characteristics of the documents and the testing sample was used to predict and verify next the efficiency of our classifier in the prediction. The data set was randomly partitioned into training (75%) and test (25%).

In our analysis, we employed 5_fold cross_validation and the effort was put on optimizing F1 which is computed with respect to very negative (VN) (which is the target class):

$$F1 = 2 * \frac{P * R}{P + R},\tag{2}$$

where *P* and *R* are defined as follows:

$$P = \frac{TP}{TP + FP},\tag{3}$$

$$R = \frac{TP}{TP + FN},\tag{4}$$

where *TP* stands for true positive, *FP* is false positive, and *FN* is false negative.

*4.4. Results*

Tables 4–8 show the polarity classification results obtained by SVM, NB, and DT classifiers for all datasets. These three classifiers were fed with all linguistic features, taken individually and by means of their combinations. The final scores were computed using Precision (P), Recall (R) and F1 scores for *very negative* class (VN).

**Table 4.** Polarity classification results obtained by SVM, NB, and DT classifiers for the Book dataset with all linguistic features taken alone and combined, in terms of P, R and F1 scores for *very negative* class (VN). The best F1 is highlighted (in bold).

| Book Features | SVM | | | Naive Bayes | | | Decision Tree | | |
|---|---|---|---|---|---|---|---|---|---|
| | **P** | **R** | **F1** | **P** | **R** | **F1** | **P** | **R** | **F1** |
| 1gTF-IDF | 0.62 | 0.34 | 0.44 | 0.33 | 0.18 | 0.23 | 0.46 | 0.36 | 0.40 |
| 1gCountVector | 0.55 | 0.51 | **0.53** | 0.34 | 0.20 | 0.25 | 0.41 | 0.38 | 0.39 |
| 1g2gTF-IDF | 0.68 | 0.34 | 0.45 | 0.43 | 0.14 | 0.21 | 0.43 | 0.36 | 0.39 |
| 1g2gCountVector | 0.57 | 0.49 | **0.53** | 0.45 | 0.15 | 0.23 | 0.43 | 0.41 | 0.42 |
| Doc2Vec | 0.57 | 0.32 | 0.41 | 0.46 | 0.62 | **0.53** | 0.40 | 0.40 | 0.40 |
| Lexicon | 0.81 | 0.18 | 0.29 | 0.51 | 0.27 | 0.35 | 0.42 | 0.38 | 0.40 |
| Doc2Vec + Lexicon | 0.64 | 0.44 | 0.52 | 0.61 | 0.45 | 0.52 | 0.42 | 0.42 | 0.42 |
| 1gTF-IDF + Doc2Vec | 0.63 | 0.49 | 0.55 | 0.34 | 0.18 | 0.23 | 0.45 | 0.42 | 0.43 |
| 1gTF-IDF + Lexicon | 0.67 | 0.41 | 0.51 | 0.35 | 0.18 | 0.24 | 0.47 | 0.40 | 0.43 |
| 1gTF-IDF + Doc2Vec + Lexicon | 0.64 | 0.51 | **0.57** | 0.35 | 0.18 | 0.24 | 0.47 | 0.43 | 0.45 |
| 1gCountVector + Doc2Vec | 0.56 | 0.52 | 0.54 | 0.34 | 0.20 | 0.25 | 0.43 | 0.40 | 0.42 |
| 1gCountVector + Lexicon | 0.59 | 0.51 | 0.55 | 0.34 | 0.20 | 0.25 | 0.53 | 0.42 | 0.47 |
| 1gCountVector + Doc2Vec + Lexicon | 0.59 | 0.51 | 0.55 | 0.34 | 0.20 | 0.25 | 0.47 | 0.43 | 0.45 |
| 1g2gTF-IDF + Doc2Vec | 0.63 | 0.49 | 0.55 | 0.44 | 0.14 | 0.21 | 0.44 | 0.39 | 0.41 |
| 1g2gTF-IDF + Lexicon | 0.69 | 0.38 | 0.49 | 0.47 | 0.14 | 0.21 | 0.48 | 0.43 | 0.46 |
| 1g2gTF-IDF + Doc2Vec + Lexicon | 0.64 | 0.51 | 0.56 | 0.47 | 0.14 | 0.21 | 0.48 | 0.40 | 0.44 |
| 1g2gCountVector + Doc2Vec | 0.58 | 0.51 | 0.54 | 0.45 | 0.15 | 0.23 | 0.45 | 0.47 | 0.46 |
| 1g2gCountVector + Lexicon | 0.58 | 0.49 | 0.53 | 0.45 | 0.15 | 0.23 | 0.46 | 0.43 | 0.45 |
| 1g2gCountVector + Doc2Vec + Lexicon | 0.60 | 0.54 | **0.57** | 0.45 | 0.15 | 0.23 | 0.48 | 0.43 | 0.45 |

**Table 5.** Polarity classification results obtained by SVM, NB, and DT classifiers for a DVD dataset with all linguistic features, taken alone and combined, in terms of P, R and F1 scores for *very negative* class (VN). The best F1 is highlighted (in bold).

| DVD Features | SVM | | | Naive Bayes | | | Decision Tree | | |
|---|---|---|---|---|---|---|---|---|---|
| | P | R | F1 | P | R | F1 | P | R | F1 |
| 1gTF-IDF | 0.74 | 0.35 | 0.47 | 0.37 | 0.17 | 0.24 | 0.54 | 0.47 | 0.50 |
| 1gCountVector | 0.56 | 0.51 | 0.53 | 0.37 | 0.17 | 0.24 | 0.47 | 0.40 | 0.43 |
| 1g2gTF-IDF | 0.70 | 0.33 | 0.45 | 0.50 | 0.11 | 0.18 | 0.48 | 0.46 | 0.47 |
| 1g2gCountVector | 0.56 | 0.49 | 0.52 | 0.50 | 0.11 | 0.18 | 0.48 | 0.40 | 0.44 |
| Doc2Vec | 0.67 | 0.30 | 0.42 | 0.33 | 0.81 | 0.47 | 0.36 | 0.41 | 0.38 |
| Lexicon | 0.69 | 0.27 | 0.38 | 0.49 | 0.57 | 0.53 | 0.46 | 0.45 | 0.45 |
| Doc2Vec + Lexicon | 0.72 | 0.49 | **0.58** | 0.34 | 0.82 | 0.48 | 0.47 | 0.47 | 0.47 |
| 1gTF-IDF + Doc2Vec | 0.74 | 0.48 | 0.58 | 0.37 | 0.17 | 0.24 | 0.56 | 0.45 | 0.50 |
| 1gTF-IDF + Lexicon | 0.72 | 0.43 | 0.54 | 0.37 | 0.17 | 0.23 | 0.53 | 0.50 | 0.51 |
| 1gTF-IDF + Doc2Vec + Lexicon | 0.69 | 0.52 | 0.59 | 0.37 | 0.17 | 0.23 | 0.48 | 0.45 | 0.46 |
| 1gCountVector + Doc2Vec | 0.59 | 0.55 | 0.57 | 0.37 | 0.17 | 0.24 | 0.48 | 0.48 | 0.48 |
| 1gCountVector + Lexicon | 0.59 | 0.53 | 0.56 | 0.37 | 0.17 | 0.24 | 0.46 | 0.40 | 0.43 |
| 1gCountVector + Doc2Vec + Lexicon | 0.62 | 0.57 | 0.59 | 0.37 | 0.17 | 0.24 | 0.51 | 0.47 | 0.49 |
| 1g2gTF-IDF + Doc2Vec | 0.72 | 0.50 | 0.59 | 0.50 | 0.11 | 0.18 | 0.53 | 0.44 | 0.48 |
| 1g2gTF-IDF + Lexicon | 0.73 | 0.45 | 0.56 | 0.47 | 0.10 | 0.16 | 0.49 | 0.46 | 0.47 |
| 1g2gTF-IDF + Doc2Vec + Lexicon | 0.71 | 0.52 | **0.60** | 0.47 | 0.10 | 0.16 | 0.51 | 0.42 | 0.46 |
| 1g2gCountVector + Doc2Vec | 0.61 | 0.57 | 0.59 | 0.50 | 0.11 | 0.18 | 0.44 | 0.42 | 0.43 |
| 1g2gCountVector + Lexicon | 0.62 | 0.55 | 0.58 | 0.50 | 0.11 | 0.18 | 0.51 | 0.42 | 0.46 |
| 1g2gCountVector + Doc2Vec + Lexicon | 0.60 | 0.56 | 0.58 | 0.50 | 0.11 | 0.18 | 0.49 | 0.45 | 0.47 |

**Table 6.** Polarity classification results obtained by SVM, NB, and DT classifiers for Electronic dataset with all linguistic features, taken alone and combined, in terms of P, R and F1 scores for *very negative* class (VN). The best F1 is highlighted (in bold).

| Electronic Features | SVM | | | Naive Bayes | | | Decision Tree | | |
|---|---|---|---|---|---|---|---|---|---|
| | P | R | F1 | P | R | F1 | P | R | F1 |
| 1gTF-IDF | 0.69 | 0.57 | 0.63 | 0.49 | 0.41 | 0.45 | 0.58 | 0.58 | 0.58 |
| 1gCountVector | 0.61 | 0.60 | 0.61 | 0.50 | 0.43 | 0.46 | 0.59 | 0.55 | 0.57 |
| 1g2gTF-IDF | 0.70 | 0.56 | 0.62 | 0.58 | 0.37 | 0.45 | 0.55 | 0.51 | 0.53 |
| 1g2gCountVector | 0.62 | 0.57 | 0.59 | 0.59 | 0.40 | 0.48 | 0.55 | 0.51 | 0.53 |
| Doc2Vec | 0.72 | 0.61 | 0.66 | 0.55 | 0.35 | 0.43 | 0.48 | 0.55 | 0.51 |
| Lexicon | 0.69 | 0.42 | 0.52 | 0.58 | 0.53 | 0.55 | 0.50 | 0.50 | 0.50 |
| Doc2Vec + Lexicon | 0.71 | 0.66 | 0.68 | 0.56 | 0.35 | 0.43 | 0.47 | 0.50 | 0.49 |
| 1gTF-IDF + Doc2Vec | 0.68 | 0.67 | 0.68 | 0.50 | 0.41 | 0.45 | 0.53 | 0.50 | 0.52 |
| 1gTF-IDF + Lexicon | 0.68 | 0.60 | 0.64 | 0.51 | 0.39 | 0.45 | 0.57 | 0.54 | 0.55 |
| 1gTF-IDF + Doc2Vec + Lexicon | 0.73 | 0.66 | **0.69** | 0.51 | 0.39 | 0.45 | 0.59 | 0.51 | 0.55 |
| 1gCountVector + Doc2Vec | 0.64 | 0.62 | 0.63 | 0.50 | 0.43 | 0.46 | 0.55 | 0.51 | 0.53 |
| 1gCountVector + Lexicon | 0.63 | 0.61 | 0.62 | 0.50 | 0.43 | 0.46 | 0.57 | 0.47 | 0.52 |
| 1gCountVector + Doc2Vec + Lexicon | 0.66 | 0.62 | 0.64 | 0.50 | 0.43 | 0.46 | 0.59 | 0.51 | 0.55 |
| 1g2gTF-IDF + Doc2Vec | 0.76 | 0.61 | 0.68 | 0.58 | 0.37 | 0.45 | 0.53 | 0.52 | 0.52 |
| 1g2gTF-IDF + Lexicon | 0.70 | 0.61 | 0.65 | 0.58 | 0.37 | 0.45 | 0.60 | 0.59 | 0.59 |
| 1g2gTF-IDF + Doc2Vec + Lexicon | 0.69 | 0.69 | **0.69** | 0.58 | 0.37 | 0.45 | 0.67 | 0.59 | 0.63 |
| 1g2gCountVector + Doc2Vec | 0.66 | 0.58 | 0.62 | 0.59 | 0.40 | 0.48 | 0.51 | 0.46 | 0.49 |
| 1g2gCountVector + Lexicon | 0.65 | 0.59 | 0.62 | 0.59 | 0.40 | 0.48 | 0.54 | 0.49 | 0.51 |
| 1g2gCountVector + Doc2Vec + Lexicon | 0.64 | 0.63 | 0.64 | 0.59 | 0.40 | 0.48 | 0.64 | 0.50 | 0.56 |

**Table 7.** Polarity classification results obtained by SVM, NB, and DT classifiers for Kitchen dataset with all linguistic features, taken alone and combined, in terms of P, R and F1 scores for *very negative* class (VN). The best F1 is highlighted (in bold).

| Kitchen Features | SVM | | | Naive Bayes | | | Decision Tree | | |
|---|---|---|---|---|---|---|---|---|---|
| | P | R | F1 | P | R | F1 | P | R | F1 |
| 1gTF-IDF | 0.71 | 0.55 | **0.62** | 0.47 | 0.45 | 0.46 | 0.59 | 0.59 | 0.59 |
| 1gCountVector | 0.64 | 0.54 | 0.58 | 0.45 | 0.49 | 0.47 | 0.57 | 0.54 | 0.55 |
| 1g2gTF-IDF | 0.70 | 0.55 | 0.62 | 0.57 | 0.39 | 0.47 | 0.59 | 0.49 | 0.53 |
| 1g2gCountVector | 0.66 | 0.52 | 0.58 | 0.56 | 0.43 | 0.49 | 0.59 | 0.51 | 0.54 |
| Doc2Vec | 0.60 | 0.36 | 0.45 | 0.45 | 0.75 | 0.57 | 0.45 | 0.49 | 0.47 |
| Lexicon | 0.60 | 0.36 | 0.45 | 0.53 | 0.64 | 0.58 | 0.48 | 0.50 | 0.49 |
| Doc2Vec + Lexicon | 0.67 | 0.57 | 0.61 | 0.50 | 0.74 | 0.59 | 0.56 | 0.54 | 0.55 |
| 1gTF-IDF + Doc2Vec | 0.75 | 0.59 | 0.66 | 0.49 | 0.45 | 0.47 | 0.51 | 0.46 | 0.48 |
| 1gTF-IDF + Lexicon | 0.66 | 0.54 | 0.59 | 0.52 | 0.44 | 0.48 | 0.57 | 0.52 | 0.54 |
| 1gTF-IDF + Doc2Vec + Lexicon | 0.73 | 0.65 | 0.69 | 0.52 | 0.44 | 0.48 | 0.55 | 0.51 | 0.53 |
| 1gCountVector + Doc2Vec | 0.72 | 0.60 | 0.65 | 0.45 | 0.49 | 0.47 | 0.55 | 0.52 | 0.54 |
| 1gCountVector + Lexicon | 0.68 | 0.57 | 0.62 | 0.45 | 0.49 | 0.47 | 0.58 | 0.50 | 0.54 |
| 1gCountVector + Doc2Vec + Lexicon | 0.71 | 0.60 | 0.65 | 0.45 | 0.49 | 0.47 | 0.57 | 0.55 | 0.56 |
| 1g2gTF-IDF + Doc2Vec | 0.75 | 0.62 | 0.68 | 0.57 | 0.38 | 0.45 | 0.57 | 0.56 | 0.57 |
| 1g2gTF-IDF + Lexicon | 0.67 | 0.58 | 0.62 | 0.57 | 0.37 | 0.45 | 0.57 | 0.51 | 0.54 |
| 1g2gTF-IDF + Doc2Vec + Lexicon | 0.75 | 0.67 | **0.71** | 0.57 | 0.37 | 0.45 | 0.60 | 0.55 | 0.58 |
| 1g2gCountVector + Doc2Vec | 0.72 | 0.60 | 0.65 | 0.56 | 0.43 | 0.49 | 0.57 | 0.52 | 0.54 |
| 1g2gCountVector + Lexicon | 0.66 | 0.57 | 0.61 | 0.56 | 0.43 | 0.49 | 0.59 | 0.51 | 0.54 |
| 1g2gCountVector + Doc2Vec + Lexicon | 0.71 | 0.61 | 0.66 | 0.56 | 0.43 | 0.49 | 0.59 | 0.56 | 0.57 |

**Table 8.** Polarity classification results obtained by SVM, NB, and DT classifiers for the Hotel dataset with all linguistic features, taken alone and combined, in terms of P, R and F1 scores for *very negative* class (VN). The best F1 is highlighted (in bold).

| Hotel Features | SVM | | | Naive Bayes | | | Decision Tree | | |
|---|---|---|---|---|---|---|---|---|---|
| | P | R | F1 | P | R | F1 | P | R | F1 |
| 1gTF-IDF | 0.75 | 0.64 | **0.69** | 0.29 | 0.73 | 0.42 | 0.64 | 0.62 | 0.63 |
| 1gCountVector | 0.67 | 0.66 | 0.66 | 0.30 | 0.69 | 0.42 | 0.66 | 0.63 | 0.65 |
| 1g2gTF-IDF | 0.77 | 0.62 | **0.69** | 0.46 | 0.40 | 0.43 | 0.62 | 0.60 | 0.61 |
| 1g2gCountVector | 0.69 | 0.65 | 0.67 | 0.50 | 0.38 | 0.43 | 0.63 | 0.57 | 0.60 |
| Doc2Vec | 0.76 | 0.47 | 0.58 | 0.40 | 0.42 | 0.41 | 0.37 | 0.43 | 0.40 |
| Lexicon | 0.69 | 0.45 | 0.55 | 0.57 | 0.70 | 0.63 | 0.51 | 0.53 | 0.52 |
| Doc2Vec + Lexicon | 0.77 | 0.62 | **0.69** | 0.56 | 0.62 | 0.59 | 0.59 | 0.61 | 0.60 |
| 1gTF-IDF + Doc2Vec | 0.77 | 0.70 | 0.73 | 0.30 | 0.73 | 0.43 | 0.65 | 0.64 | 0.65 |
| 1gTF-IDF + Lexicon | 0.79 | 0.68 | 0.73 | 0.32 | 0.70 | 0.44 | 0.68 | 0.64 | 0.66 |
| 1gTF-IDF + Doc2Vec + Lexicon | 0.78 | 0.72 | 0.75 | 0.32 | 0.70 | 0.44 | 0.69 | 0.68 | 0.69 |
| 1gCountVector + Doc2Vec | 0.72 | 0.70 | 0.71 | 0.28 | 0.75 | 0.41 | 0.67 | 0.62 | 0.65 |
| 1gCountVector + Lexicon | 0.69 | 0.69 | 0.69 | 0.29 | 0.67 | 0.41 | 0.66 | 0.65 | 0.65 |
| 1gCountVector + Doc2Vec + Lexicon | 0.74 | 0.71 | 0.73 | 0.28 | 0.75 | 0.41 | 0.69 | 0.69 | 0.69 |
| 1g2gTF-IDF + Doc2Vec | 0.79 | 0.69 | 0.74 | 0.46 | 0.40 | 0.43 | 0.68 | 0.57 | 0.62 |
| 1g2gTF-IDF + Lexicon | 0.79 | 0.68 | 0.73 | 0.46 | 0.40 | 0.43 | 0.66 | 0.61 | 0.63 |
| 1g2gTF-IDF + Doc2Vec + Lexicon | 0.81 | 0.72 | **0.76** | 0.46 | 0.40 | 0.43 | 0.71 | 0.60 | 0.65 |
| 1g2gCountVector + Doc2Vec | 0.75 | 0.70 | 0.73 | 0.50 | 0.38 | 0.43 | 0.62 | 0.59 | 0.61 |
| 1g2gCountVector + Lexicon | 0.78 | 0.65 | 0.71 | 0.50 | 0.38 | 0.43 | 0.67 | 0.61 | 0.64 |
| 1g2gCountVector + Doc2Vec + Lexicon | 0.71 | 0.68 | 0.69 | 0.50 | 0.38 | 0.43 | 0.71 | 0.60 | 0.65 |

The results, which are actually quite low due to the difficulty of the task, show that SVM is by far the best classifier for searching for the most negative opinions. SVM achieves the highest F1 scores in almost all tests. Figures 1 and 2 shows how SVM outperforms the other classifiers with all features and almost all combinations of features by computing the average of all F1 values across the four datasets. Only NB outperforms the other classifiers in one particular feature combination.

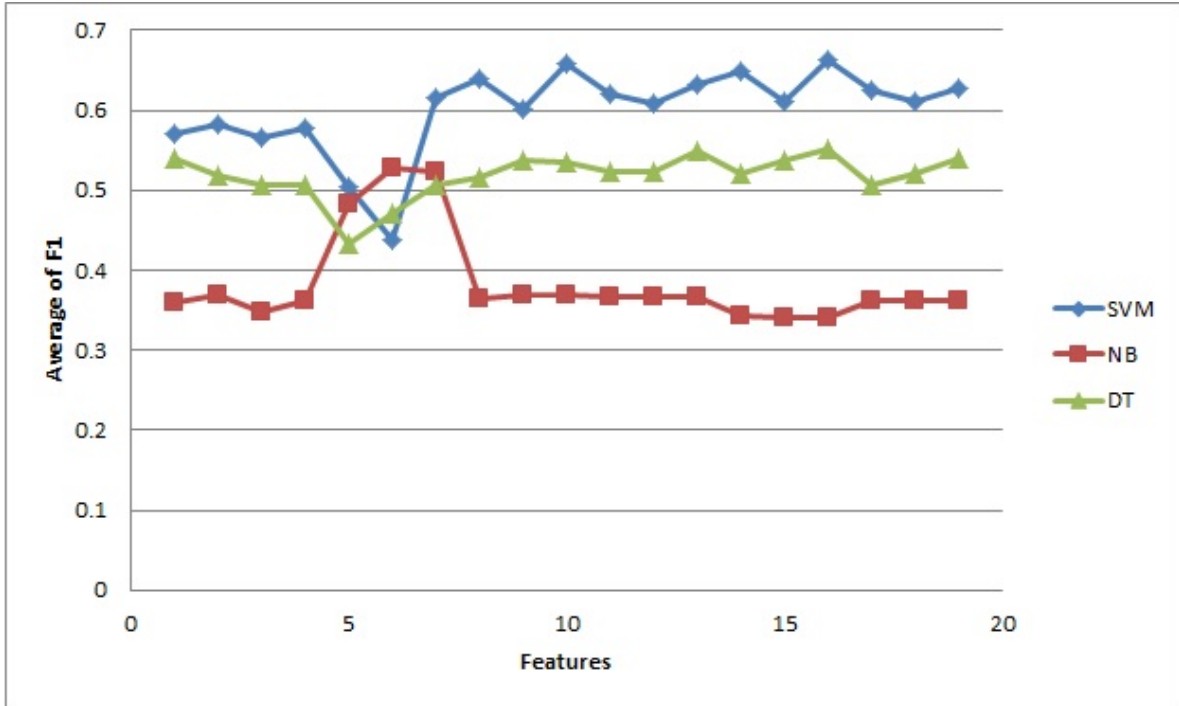

**Figure 1.** Results obtained by all classifiers for all collections with all features alone and after being combined, by computing the average of all F1 for *very negative* class (VN).

The performance of NB differs greatly depending on the number of features used in classification (see Figure 3). NB works better with a small number of features, more precisely the best scores are achieved when it only uses either Lexicon or Doc2Vec. It is worth noting that the combination of heterogeneous features hurts the performance of this type of classifier.

The DT classifier (Figure 4) has a similar behaviour to SVM (Figure 2) in terms of stability, but its performance tends to be much lower than that of SVM, as can be seen in Figure 1.

Concerning the linguistic features, the best performance of SVM (and thus of all classifiers) is reached when combining TF-IDF, whether 1g or 2g, with Lexicon and Doc2Vec, as shown in Figure 2. Thus, the combination of all feature types (n-grams, embeddings and sentiment lexicon) gives rise to the best results in our experiments. These results must be evaluated taking into account the enormous difficulty of overcoming basic features such as n-grams, which are considered as a strong baseline in tasks related to document-based classification.

Moreover, it should also be noted that the combination of just the lexicon and Doc2Vec (Doc2Vec+Lexicon) works very well with SVM and DT. This specific combination clearly outperforms the results obtained by just using either Lexicon or Doc2Vec alone, and even tends to perform better than using just n-grams, which is considered a very strong baseline in this type of classification task.

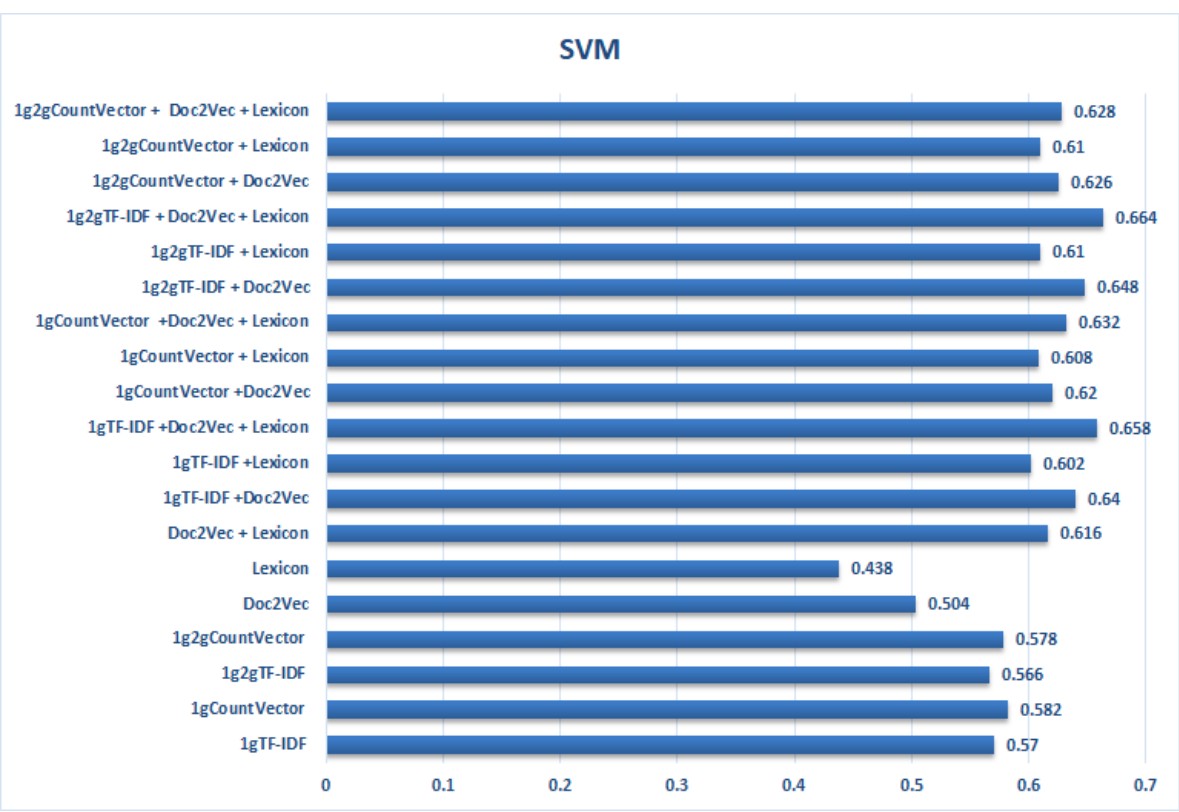

**Figure 2.** Results obtained by the SVM classifier for all collections with all features, taken individually and after being combined, by computing the average of all F1 for *very negative* class (VN).

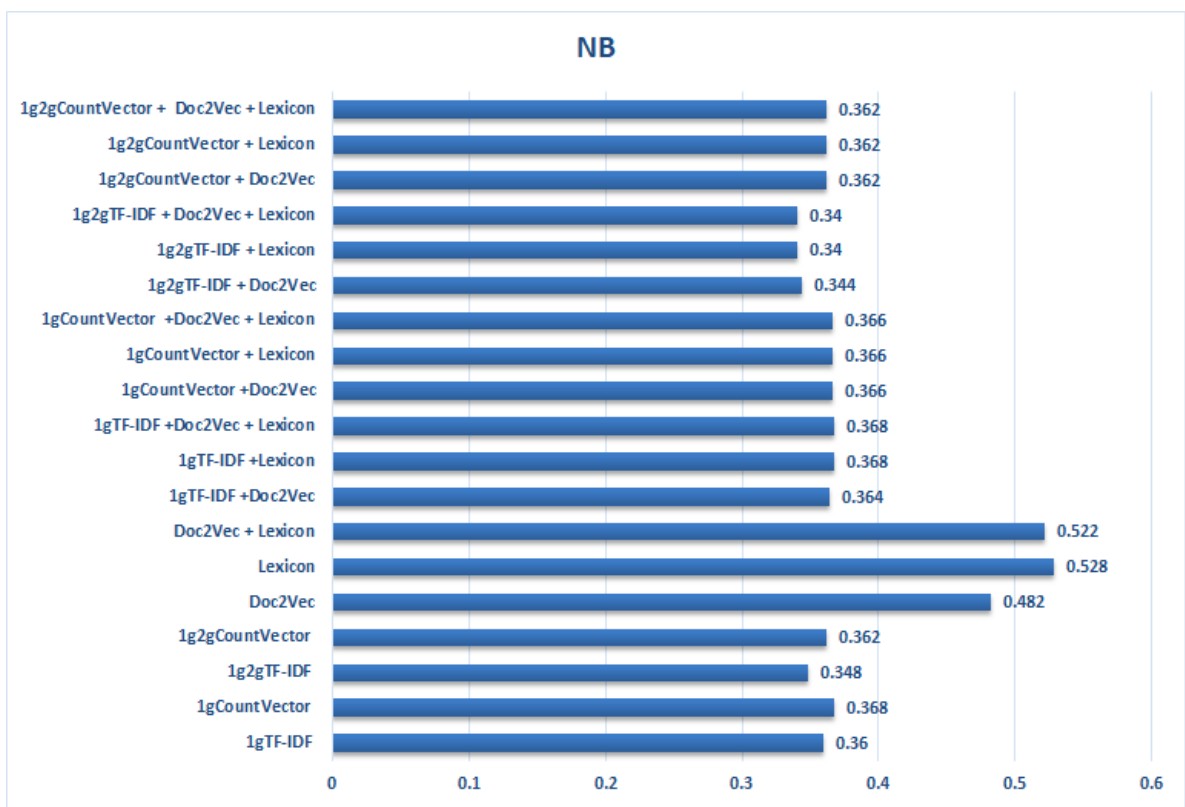

**Figure 3.** Results obtained by the NB classifier for all collections with all features, taken individually and after being combined, by computing the average of all F1 for *very negative* class (VN).

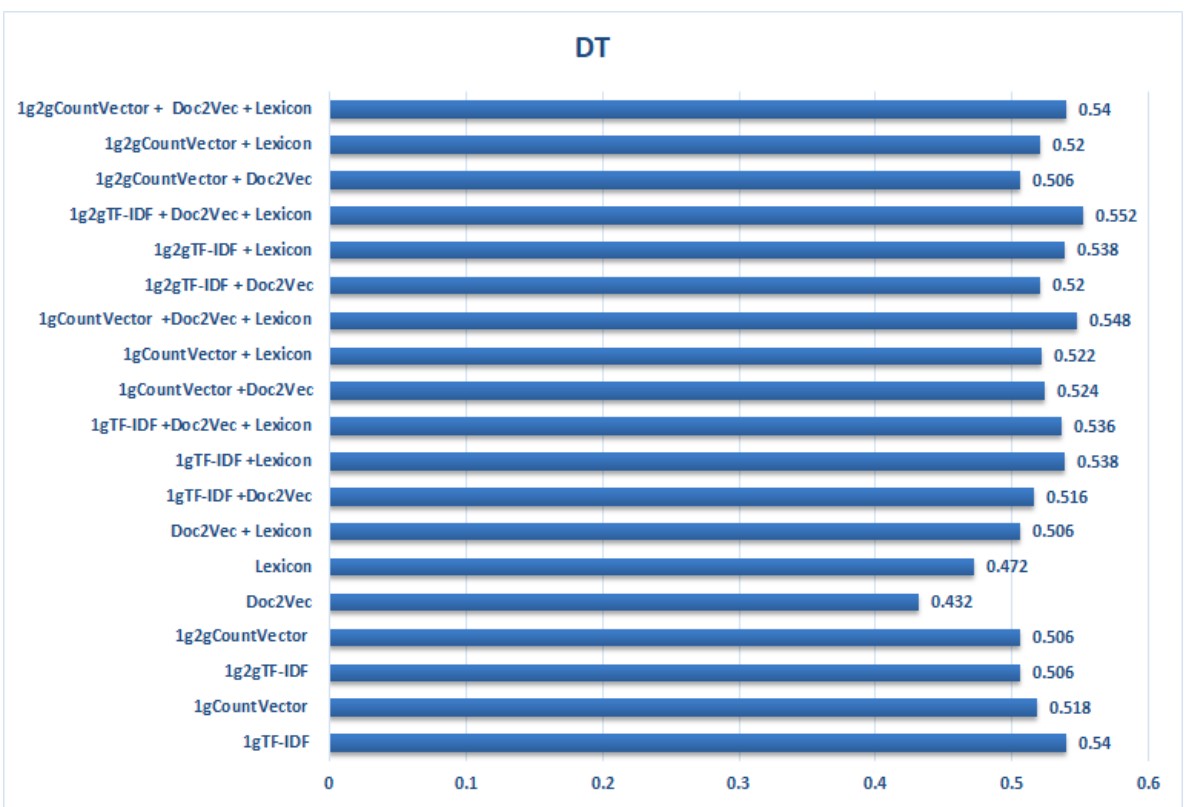

**Figure 4.** Results obtained by the DT classifier for all collections with all features taken individually and after being combined, by computing the average of all F1 for *very negative* class (VN).

## 5. Conclusions

In this article, we have studied different linguistic features for a particular task in Sentiment Analysis. More precisely, we examined the performance of these features within supervised learning methods (using SVM, NB, DT), to identify the most negative documents on four domains' review datasets.

The experiments reported in our work shows that the evaluation values for identifying the most negative class are low. This can be partially explained by the difficulty of the task, since the difference between very negative and not very negative is a subjective continuum without clearly defined edges. The borderline between very negative and not very negative is still more difficult to find than that discriminating between positive and negative opinions, since there are a quite clear space of neutral/objective sentiments between the two opinions. However, there is not such an intermediate space between *very* and *not very*.

Concerning the comparison between machine learning strategies in this particular task, Support Vector Machine clearly outperforms Naive Bayes and Decision Trees in all datasets and considering all features and their combinations.

In future work, we will compare SVM against other classifiers with the same linguistic features by taking into account not only very negative opinions, but also very positive ones (i.e., extreme opinions).

**Author Contributions:** Conceptualization, S.A. and P.G.; Methodology, S.A. and P.G.; Software, S.A.; Validation, S.A.; Formal Analysis, S.A. and P.G.; Investigation, S.A.; Resources, S.A.; Data Curation, S.A.; Writing—Original Draft Preparation, S.A.; Writing—Review & Editing, S.A. and P.G.; Visualization, S.A. and P.G.; Supervision, P.G.; Project Administration, S.A. and P.G.; Funding Acquisition, P.G."

**Funding:** This research was funded by project TelePares (MINECO, ref:FFI2014-51978-C2-1-R), and the Consellería de Cultura, Educación e Ordenación Universitaria (accreditation 2016-2019, ED431G/08) and the European Regional Development Fund (ERDF).

**Conflicts of Interest:** The author declares no conflict of interest.

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
