# Peer review of "Comparing Supervised Machine Learning Strategies and Linguistic Features to Search for Very Negative Opinions"

_information, doi:10.3390/info10010016_

Round 1
Reviewer 1 Report
In this paper, the authors studied the performance of combinations of different linguistic features and machine learning methods in discovering negative opinions. The results are interesting. However, there are still some concerns as follows:
1. Firstly and most importantly, the authors should use more words to emphasize their own contribution. The authors adopt different methods and dataset from previous work without emphasizing their own contribution.
2. More experiments on different datasets should be added. It is not sure that whether the results are representative considering this is an empirical study and there is only one dataset. It would be more convincing if the authors can provide experiments on more datasets. The significance of this paper is suppressed without evidence that the results are widely applicable.
3. The abstract should be significantly improved. As similar as the first concern, the authors briefly described the content without emphasizing their contribution.
There are some minor issues as follows:
1. In Table 4, "BOOk" should be "Book".
2. Figure 1 is hard to read. I suggest the authors to use different shape besides color to denote different methods. It might be better to separate Figure 1 to different small figures.
To sum up, the main issue of this paper is that authors failed to describe their main contribution. To improve the paper quality, my suggestions are as follows:
1. authors emphasize their theoretical contribution in theoretical aspect by introducing the differences of their methods with previous methods in the method section;
2.
authors emphasize their experimental contribution by adding more
experiments on different datasets and show that their results are widely
applicable.
Author Response
First of all, thank you very much for your useful comments about our article.
We carefully considered your comments. Herein, we explain how we revised the paper based on those comments and recommendations. We want to extend our appreciation for taking the time and effort necessary to provide such insightful guidance.
The revision, based on the review team’s collective input, includes some positive changes.
Based on your guidance:
Comment: 1. Firstly and most importantly, the authors should use more words to emphasize their own contribution. The authors adopt different methods and dataset from previous work without emphasizing their own contribution.
Response: The main objective of this article is to examine the effectiveness and limitations of different linguistic features and supervised sentiment classifiers to identify the most negative opinions in four domains reviews. It is an expanded version of a conference paper presented at KESW 2017 [10].
Our main contribution is to report on a broad set of experiments aimed at evaluating the effectiveness of different linguistic features in a supervised classification task. In addition, we also compare some supervised classifiers, namely Support Vector Machine (SVM), Naive Bayes (NB), and Decision Tree (DT), for a binary classification task consisting of searching for very negative vs. not very negative opinions.
We add a paragraph to emphasize their own contribution. (p2. lines 51 - 56).
Comment 2: More experiments on different datasets should be added.
It is not sure that whether the results are representative considering this is an empirical study and there is only one dataset. It would be more convincing if the authors can provide experiments on more datasets. The significance of this paper is suppressed without evidence that the results are widely applicable.
Response:
We conducted more experiments on a new dataset for hotel reviews in the same way of the previous experiments.
The results for new experiments have presented in (Table 8) (page 10).
Comment 3: The abstract should be significantly improved. As similar as the first concern, the authors briefly described the content without emphasizing their contribution.
There are some minor issues as follows:
1. In Table 4, "BOOk" should be "Book".
2. Figure 1 is hard to read. I suggest the authors to use different shape besides color to denote different methods. It might be better to separate Figure 1 to different small figures.
Response:
1- The abstract has been extended.
2- All typos identified by the reviewers (and other ones) have been corrected.
3- the figures have been separated. (Figs. 1 to 4)
Reviewer 2 Report
The paper investigates how several machine learning strategies and linguistic features can be used to discover very negative opinions. The topic is interesting and definitely worth investigating.
#1 Abstract
The abstract can be considered too short and missing vital information concerning the results obtained in the paper. The authors are advised to also include a few comments about the results that they have achieved. A short phrase pointing out why detecting highly negative opinions is important / useful should also be included
#1. Introduction
The introduction states the contributions of the paper. It also provides an adequate amount of information regarding the topic.
#2. RelatedWork
The related works section could be improved by including more references for the statements made. For example, at line 54, the authors state that there are two main approaches for performing sentiment analysis. It would be useful for the readers of the paper if the authors included at least one-two references for each approach.
# Additional suggestions
- the text in Figure 1 is relatively hard to read. The authors are advised to improve its quality. The figure should also have a label for the vertical axes (most likely de F1 score).
Author Response
First of all, thank you very much for your useful comments about our article.
We carefully considered your comments. Herein, we explain how we revised the paper based on those comments and recommendations. We want to extend our appreciation for taking the time and effort necessary to provide such insightful guidance.
The revision, based on the review team’s collective input, includes some positive changes.
Based on your guidance:
Comment 1: The abstract can be considered too short and missing vital information concerning the results obtained in the paper. The authors are advised to also include a few comments about the results that they have achieved. A short phrase pointing out why detecting highly negative opinions is important/useful should also be included.
Response:
The abstract has been extended.
Comment 2: The related works section could be improved by including more references for the statements made. For example, at line 54, the authors state that there are two main approaches for performing sentiment analysis. It would be useful for the readers of the paper if the authors included at least one-two references for each approach.
Response:
Many references have been summarized in table 1 and on related work section.
- We conducted more experiments on a new dataset for hotel reviews in the same way of the previous experiments.
The results for new experiments have presented in (Table 8) (page 10).
- The written English in the revised paper has been checked by a native English speaker.
Round 2
Reviewer 1 Report
In this paper, the authors examined the performance of several classifiers in the process of searching for very negative opinions. It is very interesting to see different ML method such as SVM, NB, DT in this application.
Reviewer 2 Report
The authors have addressed the comments in the previous review.